# Strengthening of the Fe-Ni Invar Alloy Through Chromium

**DOI:** 10.3390/ma12081297

**Published:** 2019-04-20

**Authors:** Qingshuang Sui, Jun He, Xin Zhang, Zhonghua Sun, Yunfei Zhang, Yingfei Wu, Zhixiang Zhu, Qiang Zhang, Huifen Peng

**Affiliations:** 1School of Materials Science and Engineering, Hebei University of Technology, Tianjin 300130, China; suiqingshuang@163.com (Q.S.); hejun19861102@163.com (J.H.); zhang_xin@hebut.edu.cn (X.Z.); 2HBIS Group Technology Research Institute, HBIS Group, Shijiazhuang 052165, China; zhangyunfei@hbisco.com (Y.Z.); wuyingfei@126.com (Y.W.); 3State Key Laboratory of Advanced Transmission Technology (Global Energy Interconnection Research Institute Co., Ltd.), Beijing 102211, China; zhuzhixiang003@163.com (Z.Z.); 13601210056@126.com (Q.Z.); 4Tianjin Key Laboratory of laminating Fabrication & Interface Control Technology for Advanced Materials, Tianjin 300130, China

**Keywords:** invar alloy, precipitation strengthening, aging, Cr alloying

## Abstract

Invar alloys with both high strength and low thermal expansion are urgently needed in fields such as overhead power transmission, aero-molds, and so on. In this paper, Cr was introduced as a cost-efficient alloying element into the Fe-36Ni binary invar alloy to increase its mechanical strength. Our results confirmed that fine Cr_7_C_3_ precipitants, together with some Fe_3_C, in the invar alloy aged at 425 °C could be obtained with a short aging time. Those precipitants then grew and aggregated at grain or sub-grain boundaries with an increase in aging time. Simultaneously, mechanical strength and coefficient of thermal expansion (CTE) parabolically varied with the increase in aging time. The sample aged at 425 °C for 7 h presented a maximum strength of 644.4 MPa, together with a minimum coefficient of thermal expansion of 3.30 × 10^−6^ K^−1^ in the temperature range of 20–100 °C. This optimized result should be primarily attributed to the precipitation of the nanoscaled Cr_7_C_3_.

## 1. Introduction

Fe-Ni binary alloy with a Ni content of ~36 wt.% is known as invar alloy, which shows a very low coefficient of thermal expansion (CTE) of almost zero under 100 °C [1,2,3,4]. Such an anomalous thermal expansion effect of the Fe-36Ni alloy is known as the invar effect [5]. Therefore, the Fe-36Ni invar alloy was initially developed as a representative of low-expansion materials [6]. However, with the development of industrial techniques, it has become a challenge to meet the fast growing requirement of multiple industrial fields, such as original precision instruments, large electronic telescope base positioning devices, double capacity conductors, and LNG carriers, because of the bottleneck caused by the low mechanical strength of invar alloys.

To address this issue, multiple strategies have been proposed, of which precipitation hardening is a very promising strengthening method. Generally, it can be classified as intermetallic strengthening and carbide hardening for invar alloys. The intermetallic strengthening is induced by alloying elements such as Ti, Al, and Nb, which can form fine γ′-Ni_3_Al or Ni_3_ (Al, Ti) secondary phases while aging [7,8]. Note that the mechanical strength of materials can be improved at the expense of thermal expansion characteristics [9] because, in the austenite matrix, precipitation of the γ′ phase consumes Ni, which is the key element for low CTE characteristics. This drawback could be offset by introducing Co as an alloying element, which will unfortunately increase production costs.

Compared with intermetallic strengthening, carbide hardening is more cost-efficient, in which precipitation of fine carbides such as Mo_2_C, TiC, and VC greatly improves the strength of invar alloys without deterioration of their low expansion characteristics. Generally, fine carbides could result in improved strengthening effects. However, this type of very fine carbides can only be obtained by solution treatment and subsequent aging. It is worth noting that the melting points of carbides such as Mo_2_C, VC, and TiC are known to be as high as ~2500 °C [10]. Accordingly, a high solution treatment temperature >1200 °C is required to dissolve the added alloying elements and their carbides into the austenite as much as possible. In order to obtain a high mechanical strength, Ha et al. [11] revealed that solutionizing at 1280 °C was necessary to dissolve the Mo-enriched coarse precipitates into the austenite matrix for the Fe-36Ni-2.75Mo-0.9Co-0.7Cr-0.32C-0.17Si-0.23Mn invar alloy. This high temperature process is always concomitant with an abnormal growth in austenite grain and serious oxidation of materials. To offset those drawbacks, we recently used a lower solutionizing temperature at 1050 °C for the Mo-strengthened invar alloy [12]. It is probably an uncompleted dissolution of the primary Mo_2_C carbides that results in an unideal precipitation strengthening effect, and only a maximum strength of 820 MPa was obtained for the studied invar alloy. Therefore, alloying elements, which possess good precipitation strengthening and form carbides with a little lower melting point, are highly desired.

Unlike the Mo, V, and Ti alloying elements, Cr is a cheap and widely used carbide-forming element as its carbides can dissolve into austenite completely at temperatures <1100 °C [10,13,14]. Accordingly, as an alloying element, Cr can facilitate solution treatment of invar alloy, and then is expected to produce a pronounced precipitation strengthening effect by forming high-hardness carbides such as Cr_23_C_6_ or Cr_7_C_3_ with subsequent aging. Yu et al. [15] found that adding Cr could increase the hardness of Fe-36Ni invar alloy by grain refinement. However, a maximum hardness of HV 170 was obtained for the studied alloy, accompanied with a rise in the CTE value. On the other hand, Nakama et al. [3] tried to substitute the V in the 0.2C-36Ni-0.8V invar alloy with Cr and Ti in order to reduce the consumption of rare metals. They found that 0.6 wt.% Cr and 0.2 wt.% Ti in replacement of 0.2 wt.% V only resulted in a slight increase in tensile strength at the expense of a deterioration in thermal expansion characteristics. In this paper, Cr was added to the invar alloy of Fe-36Ni with the aim to improve its mechanical strength without affecting its thermal expansion performance. Moreover, the effect of precipitation strengthening derived from secondary phases during aging was investigated.

## 2. Experimental Procedure

### 2.1. Materials Preparation

A 50 kg ingot of Fe-33.6Ni-6.5Cr-0.19C invar alloy was smelted in a DDVIF-50-100-2.5 vacuum induction furnace (HBIS Group, Shijiazhuang, China) under an argon atmosphere. The ingot was hot-rolled into a plate having a thickness of 8 mm, cooled in air, and then was homogenized at 1250 °C for 1.5 h. All the samples for mechanical property measurements were quenched in water after solutionizing at 1000 °C for 1 h and then aged at temperatures of 375–475 °C for various times from 1 to 14 h.

### 2.2. Microstructures Characterization

To protect precipitates from peeling off, a constant voltage of 30 V was maintained between the samples and a stainless steel electrode in an electrolytic solution of glacial acetic acid and perchloric acid in a volume ratio of 4:1 at 0 °C for electrolytic polishing. Then, the samples were observed under a Zeiss ULTRA-55 FE-SEM (HBIS Group, Shijiazhuang, China), and chemical compositions of precipitates were determined by X-Max50 EDS, which was attached to the SEM. As the precipitates were small in size and less in amount, large errors were noticed for determining the specific precipitated phases for the aged samples. We electrochemically extracted the aged samples in an aqueous solution of 1 wt.% NaCl, 0.4 wt.% citric acid, 1 wt.% FeSO_4_, and 0.5 wt.% NiSO_4_ at a voltage of 5 V. During extraction, citric acid was supplemented to remain a constant pH value of 2–3 for the aqueous solution. The extracted products were separated from the solution and then cleaned several times with deionized water. After drying, the extracted products were hand-milled for 30 min in a zirconia mortar. The total weight of the extracted products was about 0.5 g. X-ray diffraction (XRD) measurements were conducted on a 4B9A Beijing Synchrotron Radiation Facility (Beijing, China) at a scanning rate of 0.02°/s with steps of 0.02° to detect phase constituents of samples at different aging times. Some extracted solution was then diluted with deionized water at a volume ratio of 1:5, and then the copper grid was used to fish for precipitates in the diluted extracted solution. After drying, the copper grid was installed on a JEOL JEM-2100 for morphology observation, EDS analysis, and selected area electron diffraction (SAED).

### 2.3. Characterization of Mechanical Properties

Mechanical properties of all samples were measured on an SHT-5305 material testing machine at a loading rate of 2 mm/min. Hardness measurements were conducted on an HXD-1000 Vickers tester (Tianjin, China) with a load of 100 g. 

### 2.4. Characterization of Thermal Expansion Properties

Thermal expansion of the samples was tested in a temperature range from −196 to 300 °C under a modified Leica J11 dilatometer, and Equation (1) was used to calculate the linear CTEs of these materials:(1)$$\bar{\alpha }=\frac{dL}{L_{0}}\cdot \frac{1}{dT}$$
where α is the CTE, dL/L_0_ is the relative variation in specimen length, L_0_ is the original length of the samples, and dT is the change in temperature.

## 3. Results

### 3.1. Mechanical Behavior of Invar Alloy under Aging

For a high-Cr die steel, an optimum strengthening effect caused by the precipitation of Cr carbides is generally obtained when tempering at 500 °C after quenching [10,13]. Given the difference in chemical compositions and microstructures between the invar alloys and the high Cr die steel, we slightly decreased the aging temperature for the current invar alloy under our experimental condition. Figure 1 shows the hardness variation of samples at different aging conditions. The fitted curves show similar characteristics, i.e., the hardness gradually increases and reaches a maximum, and then declines with further increase in either aging temperature or aging time. The maximum hardness of 286 HV, which is 63.4% higher than that of the solid solution treated one, was obtained when aging at 425 °C for 10 h. A sharp increase in the hardness of samples when aging at 375–425 °C should be attributable to the precipitation of Cr carbides from the austenite matrix. However, the further increase in aging temperature resulted in an apparent decrease in hardness, i.e. post-aging, because of the growth of precipitated Cr carbides.

Figure 2 shows the stress–strain curves for samples treated with different processes. The hot-rolled sample demonstrated a tensile strength of 543 MPa, which is ~11% higher than that of the conventional Fe-36Ni invar alloy without any other alloying elements [16]. Note that this result suggests that Cr alloying plays an important role in enhancing the strength of invar alloy. Solid solution treatment at 1000 °C resulted in a slight decrease in strength. This phenomenon is related to the dissolution of primary carbides into austenite while heating [17,18]. Note that aging at 425 °C produced an apparent increase in strength, and the sample aged for 7 h presented a maximum strength of 644.4 MPa, which is ~45% higher than that of the solid solution-treated one. Further prolongation in aging time resulted in a decrease in mechanical strength.

### 3.2. Thermal Expansion Behavior of Invar Alloy during Aging

Figure 3 shows the thermal expansion curves of specimens treated with different processes. It can be seen that the solutionization at 1000 °C led to an increase in the CTE value because of the dissolution of primary carbides in the austenite matrix. However, aging at 425 °C gave rise to a decrease in CTE again because of the precipitation of Cr carbides. It is worth noting that the invar alloy aged at 425 °C for 7 h exhibited the minimum CTE of 3.30 × 10^−6^ K^−1^ in the temperature range of 20–100 °C. This value is very close to what we reported in the single Mo-alloyed invar alloy [12].

Cr carbides are known to have a larger CTE value (10 × 10^−6^ K^−1^) compared to Mo- or Ti-carbides [19]. Contrary to our obtained results, the precipitation during aging should increase the CTE value of the material. However, Sun et al. [20,21,22] found that expansion coefficients of the MnNx-based compounds with anti-perovskite structure were directly proportional to their grain size. In fact, compounds with coarse grains demonstrated an expansion coefficient of the order of 10^−5^ K^−1^. However, this value decreased to <1% (~0.12 × 10^−6^ K^−1^) for compounds with a grain size of ~12 nm [21]. Accordingly, it was deduced that Cr carbides at a nanoscale level could be the primary precipitates, which has negligible effect on the CTE value of the short-aged invar alloy. However, further prolongation in aging time resulted in an unexpected increase in the CTE value, and the invar alloy aged for 10 h demonstrated a CTE value of 4.71 × 10^−6^ K^−1^. This is ~13.5% and 42.7% greater than that of the hot-rolled alloy and the 7 h-aged alloy, respectively. These results suggest that the CTE value was sensitive to the aging process, i.e., variation in size of the precipitated carbides.

## 4. Discussion

Figure 4 shows the SEM images of samples aged at 425 °C for different times. The invar alloy after solutionization at 1000 °C was composed of single austenite phase (Figure 4a). This result suggests that primary carbides are almost completely dissolved into the austenite in the form of solid solution. Aging at 425 °C resulted in preferential re-precipitation of secondary phases along either grain or sub-grain boundaries in the austenite, which is indicated by the small circles and white arrows in Figure 4b,c. This means that grain or sub-grain boundaries acting as crystal defects supply additional nucleation sites and accordingly result in inhomogeneous precipitation of the secondary phases. However, the secondary phases are finer along sub-grain boundaries than those along grain boundaries. It is then deduced that secondary phases precipitated in the austenite interior are probably much finer than the others. Unexpectedly, the secondary phases grow and aggregate along the grain or sub-grain boundaries with prolongation in aging time. Apparent rods of the secondary phases, indicated by black arrows in Figure 4e, can be observed for the 10 h-aged sample. These results can well explain the deterioration in mechanical strength and thermal expansion characteristics shown in Figure 2 and Figure 3. 

In order to correctly determine the secondary phases precipitated in the aged samples, we diluted the extracted solution to fish for the precipitates and then observed them under TEM. Figure 5a shows TEM morphology of the precipitates in the sample aged for 7 h. They are composed of two types of particles: a smaller spherical one (~100 nm in diameter) and a larger blocky one (~1.2 μm in length and 0.6 μm in width). They are deduced to correspond to the secondary phases precipitated in the austenite interior and along sub-grain boundaries, respectively, based on the fact that the smaller particles are difficult to resolve under SEM like those in Figure 4. Moreover, calibration to the SAED pattern shown in Figure 5b proves that the smaller particles are Cr_7_C_3_ carbide. The larger particles should also be Cr_7_C_3_ carbide due to the nearly identical EDS results for those particles shown in Figure 5d,e. The fine particles should possess a strong strengthening effect, corresponding to the sharp increase in hardness and strength shown in Figure 1 and Figure 2.

Figure 6a shows XRD patterns of the samples aged for different times. In addition to strong XRD peaks attributable to the austenite matrix, XRD peaks due to the precipitated products are difficult to observe. Differently, Figure 6b clearly indicates that the precipitated products consist of two kinds of carbides like Cr_7_C_3_ and Fe_3_C for those samples. In addition to XRD peaks due to the Cr_7_C_3_ and Fe_3_C carbides, some XRD peaks attributed to austenite appear in the extracted products for the 5 h-aged sample. Just like what we observed in Figure 4b, the precipitated products in this sample are less in amount and small in size, which causes great difficulty in completely separating the carbides with the austenite matrix. That should be the reason why some austenite presents in the extracted products of the 5 h-aged sample. Growing larger with the increase of aging time, the precipitated carbides are easy to deposit at the bottom of the container, which brings convenience for their washing and separating with the austenite matrix. Accordingly, only XRD peaks of the Cr_7_C_3_ and Fe_3_C carbides are observed, and their intensities apparently increase in the 7 h- and 10 h-aged samples. Those results are in good agreement with those obtained in Figure 4c,e.

Based on the measured chemical compositions, it is known that the mole ratio of Cr to C for the present invar alloy is about 7.9, which is about twice as large as that of the Cr_23_C_6_ carbide. Theoretically, precipitation of the Cr carbides during aging should be the Cr_23_C_6_, just like what happens in the high-Cr die steel [13]. Precipitation nucleation is a kind of diffusion transformation which relies strongly on diffusion of carbon atoms and chromium atoms. At 425 °C, the diffusion coefficient of carbon and chromium atoms is 6.7 × 10^−12^ and 1.42 × 10^−21^ cm^2^/s, respectively, in the austenite matrix for the present invar alloy [23]. The diffusion coefficient of carbon atoms is significantly higher than that of chromium atoms, so some cementite, Fe_3_C, are easy to precipitate, in addition to the Cr_7_C_3_ carbides. Similar phenomena occurred in the Ti-V-Mo complex microalloyed steel reported by Zhang et al. [24]. Because of the low Cr content of the EDS result in Figure 4d, it is deduced that the large particles precipitated along grain boundaries, which supply a rapid migration channel for carbon atoms, are the Fe_3_C cementite.

It is known that Fe_3_C has a weaker strengthening effect than special carbides like Cr_23_C_6_, VC, and Mo_2_C [3,11,12]. In order to prohibit the precipitation of Fe_3_C from the austenite matrix, it is better to suitably increase carbon and chromium contents in future research. Considering the new investigation on a Fe-Ni invar alloy reinforced by WC nanoparticles reported by Zheng et al. [25], we suggest that the solutionized invar alloy should be cold-rolled before aging to apparently increase the number of precipitation sites for the carbides. That would be favorable not only to decrease the size of the precipitated carbides and increase their amount, but also to produce little effect on their thermal expansion characteristics.

## 5. Conclusions

The effects of aging on microstructures and mechanical and physical properties were investigated for a Fe-33.6Ni-6.5Cr-0.19C invar alloy. It was revealed that Cr_7_C_3_ particles, together with some Fe_3_C, precipitated in the invar alloy during aging at 425 °C. Moreover, their sizes were in the nanoscale after a short aging time. However, they grew and aggregated at grain and sub-grain boundaries with an increase in aging time. The mechanical strength of the material parabolically varied with the increase in aging time and reached a maximum of 644.4 MPa after 7 h of aging. In contrast to the variation in strength, the CTE value presented a minimum of 3.30 × 10^−6^ K^−1^ in the temperature range of 20–100 °C after an aging time of 7 h. Any further increases in aging time would result in a decrease in mechanical strength and an unexpected increase in CTE.

## Figures and Tables

**Figure 1 materials-12-01297-f001:**
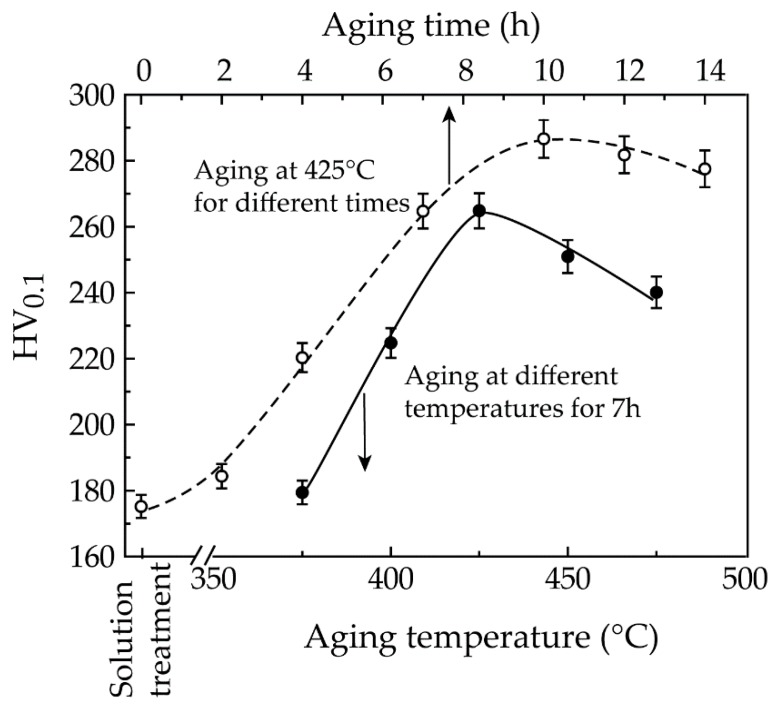
Hardness of samples treated with different processes.

**Figure 2 materials-12-01297-f002:**
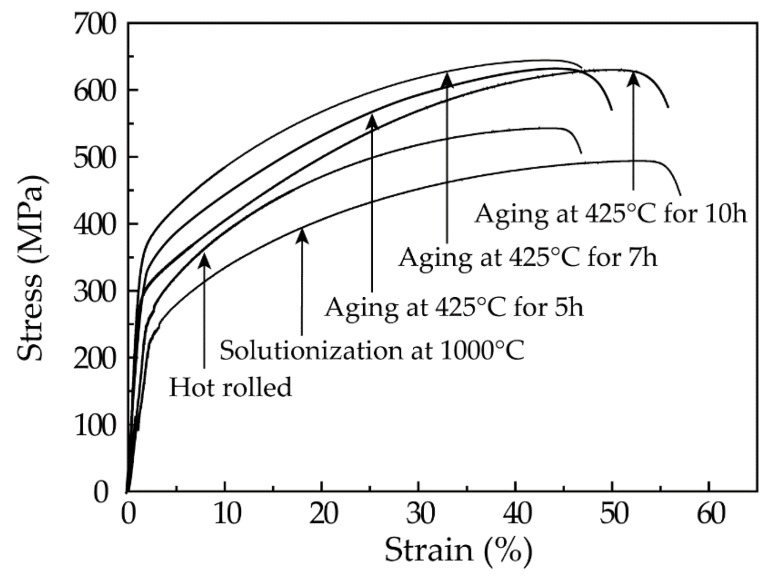
Strain-stress curves of samples treated with different processes.

**Figure 3 materials-12-01297-f003:**
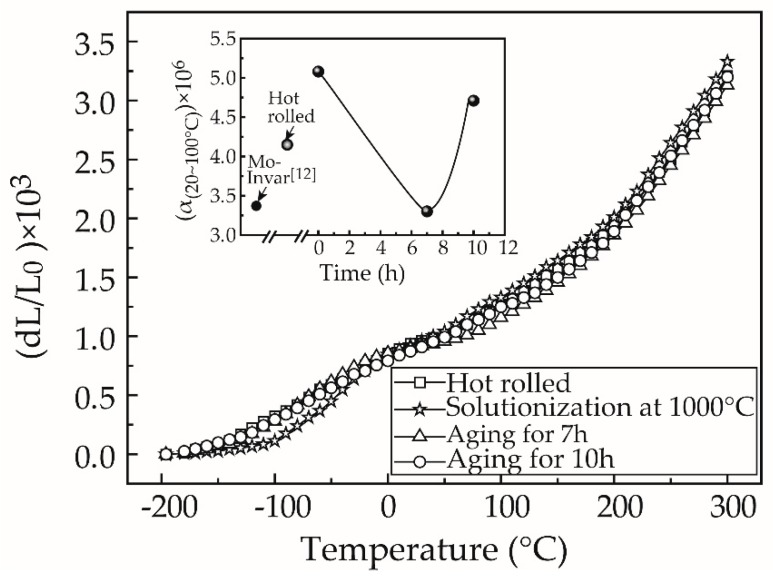
Linear thermal expansion curves of the hot-rolled invar alloy and those aged at 425 °C for different times.

**Figure 4 materials-12-01297-f004:**
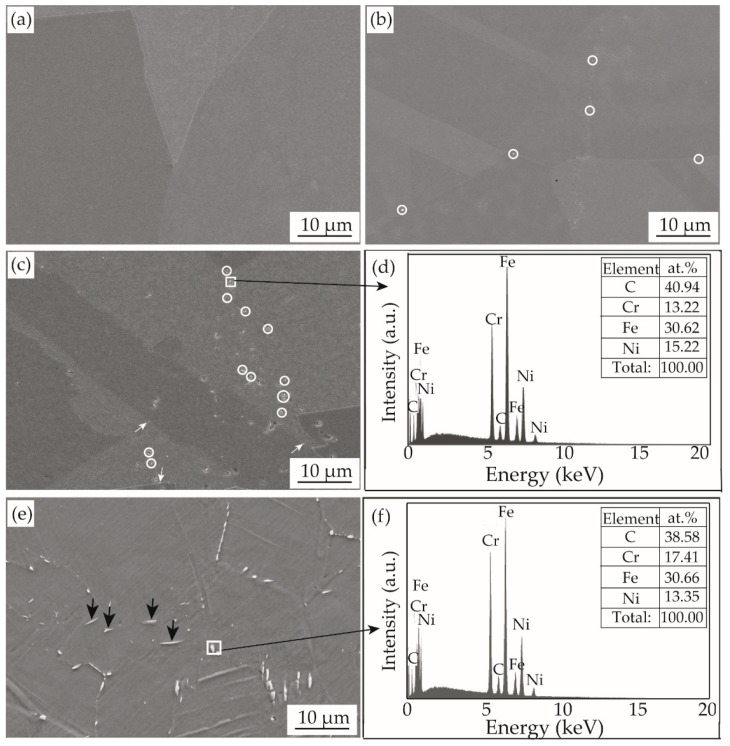
SEM images and EDS results of the invar alloy aged at 425 °C for different times: (**a**) 0 h, (**b**) 5 h, (**c**,**d**) 7 h, and (**e**,**f**) 10 h.

**Figure 5 materials-12-01297-f005:**
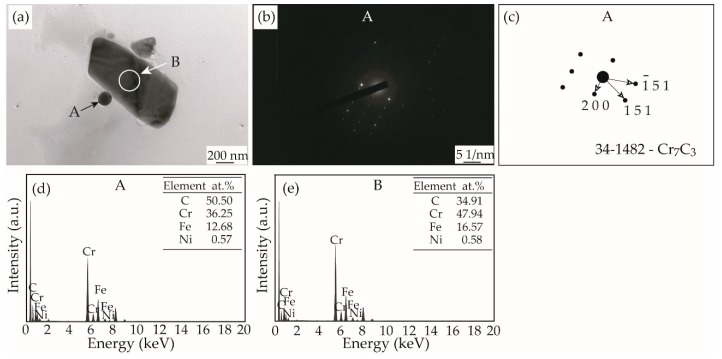
TEM morphologies of the extracted precipitates in the sample aged at 425 °C for 7 h (**a**), SAED pattern (**b**,**c**) and EDS results (**d**,**e**).

**Figure 6 materials-12-01297-f006:**
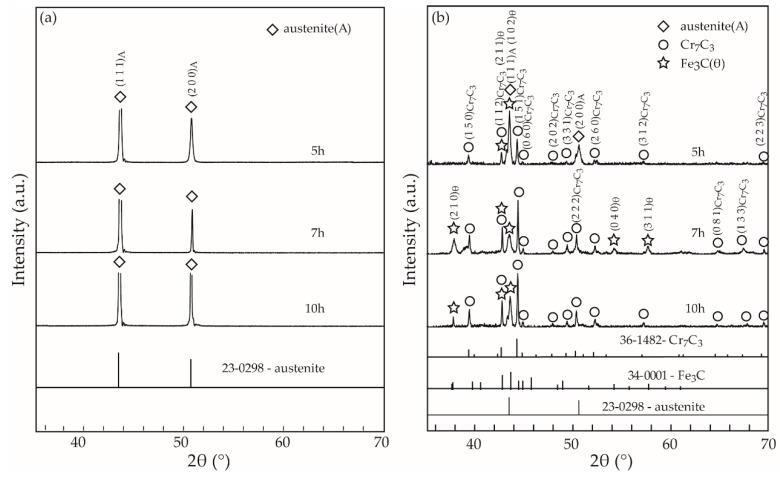
XRD patterns of the samples (**a**) and the extracted products (**b**) aged for different times.

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
