# Peer review of "Strengthening of the Fe-Ni Invar Alloy Through Chromium"

_materials, 2019, doi:10.3390/ma12081297_

Reviewer 1 Report

This paper reports the mechanical properties, thermal expansion and the microstructure of Cr added Fe-Ni invar alloy. The authors have found that a maximum strength and a minimum coefficient of thermal expansion are simultaneously achieved in the sample aged at 425 degree C for 7 h. The origin has been carefully examined by observing the microstructure and is attributed to the precipitation of the nano-sclaed Cr7C3.

The paper is well organized and has revealed that Cr is very important element to improve the performance of invar alloys, which have a drawback of low mechanical strength. So I believe the manuscript meets all criteria necessary for Materials. But, before the acceptance, I recommend the authors to address the comment listed below.

 (1)    I recommend the authors to add a brief survey of papers, reporting Cr added invar alloys. I think it is useful for the general readers.

Author Response

Answer: We supplement a brief survey on the Cr-added invar alloys reported by others in the introduction part. Please check it in the revised manuscript.

Reviewer 2 Report

In this paper, the effect of Cr on the microstructure, tensile and liner expansion of the invar alloy was investigated. The manuscript is generally well-written and can be published after addressing the following comments:

1- It is unclear how many samples were pulled for tensile testing? Uncertainty values should be provided.

2- Authors did not explain the hardness and thermal expansion measurements in the experimental procedure. It is better explain it again here rather than referring to the previous work.

3- If collecting sample was difficult for TEM observation, how authors did collect enough samples for XRD analysis? It is unclear how XRD analysis was conducted on the extracted particles. Please explain in more details.

4- What was the cooling rate after homogenising at 1250°C and solutionising at 1000°C? Air cooled? Furnace cooled? Or quenched in water and/or oil?

5- XRD patterns must be indexed.

6- There are many works in the literature regarding invar effect of FeNi alloy. It could be beneficial to readers if authors provide an image showing where this alloy is standing compared to others.

Author Response

1. Reply: We used at least three samples for tensile testing. When some abnormal results occurred, we used additional samples for further testing until obtained a stable result. Different from the hardness presented in Fig.1, it is difficult to use error bars exhibiting the stress-strain curves, so we choose a medium one as the model to exhibit the variation in stress for the samples treated under different processes.

2. Reply: We have supplemented the hardness and thermal expansion measurements in the experimental procedure in the revised manuscript. Please check the details in it.

3. Reply: From the chemical composition of the present invar alloy, it is known that the precipitated carbides are less in amount (≯3.0 wt.%). When using a bulk sample for XRD measurements, it is difficult to accurately determine which kinds of carbides they are, due to the disturbance of austenite matrix. On the other hand, TEM samples are small in size (≤φ3 mm in diameter and ≯10 μm in thickness). Furthermore, thinning is a technique hard to obtain a perfect sample used for ideal TEM observation, and fine particles are easy to fall during thinning. Accordingly, information obtained on the precipitated carbides from the TEM observation is generally incomplete. Based on the reasons above, we tried to use powder XRD measurements to analyze the precipitated products during aging.

The samples used for the powder XRD measurements are about 4 mm in thickness, 8 mm in width and 50 mm in length. Oxide layers on them were mechanically removed, and then they were submerged into an aqueous solution of 1 wt.% NaCl, 0.4 wt.% citric acid, 1 wt.% FeSO4 and 0.5 wt.% NiSO4 to proceed an electrochemical extract under a voltage of 5 V. During extracting, some citric acid was supplemented to remain a constant pH value of 2~3 for the aqueous solution. The extracted products were separated from the solution and then cleaned several times with deionized water. After drying, the extracted products were hand-milled for 30 minutes in a zirconia mortar. The total weight of the extracted products is about 0.5 g. The above information was supplemented in the Experimental section of our revised manuscript.

4. Reply: The cooling rate after homogenizing at 1250 ℃ is slow, i.e. cooling in air, whereas the cooling rate after solutionizing at 1000 ℃ is fast, i.e. quenching into water to prevent the carbides from precipitating. We supplement the above information in the revised manuscript.

5. Reply: XRD patterns in Fig.6 are already indexed, please check details in the revised manuscript.

6. Reply: We supplement the content in the Introduction relevant to the invar alloys strengthened by Cr reported by others. Please check details in the revised manuscript.

Reviewer 3 Report

The aim of the paper, i.e. the influence of Cr additive on mechanical properties of the Fe-36Ni alloy, is quite interesting, even if not so appealing, due to the large number of papers already published on invar alloy.

The abstract summarize the work. Unfortunately, the purpose of the study is not clearly outlined and the findings of prior work are not well discussed. Authors didn’t mention if there is information about modification of invar alloy by Cr additive. There are no errors in logic or experimental procedure. The authors accurately explain how the data were collected. There is sufficient information that the experiment can be reproduced. All topics are well presented and discussed. The summary and conclusions are sound and justified. All presented figures are good quality and they prove their point. The paper is written in good English. The manuscript is easily readable concerning language, style and presentation. The references are appropriate and up to date.

Author Response

Reply: We supplement the findings of prior work in the introduction section and give some suggestion on further modification of invar alloy by Cr additive in the Discussion Section. Please check the detail in the revised manuscript.

Reviewer 4 Report

This article concerns well-known Fe-36Ni invar alloy modified by carbon and chromium addition and reports detailed analysis of both, matrix and precipitations. The samples aged (425oC) with different times were mechanically (hardness measurements, tensile test) and thermomechanically (CTE measurements) tested. It is worth noting that manuscript is written in logical way. However some of experiments/results require additional description/explanations. Due to I suggest major revision as follows:

1.       The title must be strictly corresponding with the article – effect of strengthening is obtained by simultaneously addition chromium and carbon.

2.       Line 22 – If mechanical strength and CTE are parabolically varied with aging time you must proof it by correlation coefficient calculation.

3.       Line 31 – please delete the semicolon and start new sentence (check throughout the article).

4.       The manuscript novelty must be clearly given. Due to the obtained results should be compared with commercial invar alloy.

5.       Line 56 - Unlike alloying elements of Mo, V and Ti, Cr is a cheap…. – are you sure, see below? Mo (44$ per 100g), V (20$/lb), Ti (30$/lb) versus Cr (32$ per 100g).

6.       The methodology of mechanical properties and CTE measurements must be precisely, step by step described in Materials and Methods section (please use the Materials journal template).

7.       Line 66 – please use properties instead of property - check throughout the article.

8.       Line 67 - Authors reported that aging temperature was 375-475oC but in the line 94 temperature of 375-425oC is given. Moreover only results for aging at 475oC were presented in the article – please explain it.  

9.       Line 81 – the abbreviation EDS is more suitable than EDX.

10.   Line 90 – The fitted curve…. – How was the curve fitted to experimental points, the correlation coefficient is needed.

11.   Figure 4d,f and Figure 5d,e – the chemical composition should be given in at. %.

12.   Figure 5 – please use “EDS results” instead of “EDX  resultes”.

 The English requires grammatical improvement – please carefully check the article.

Author Response

1. Reply: Carbon is very common element added in ferrous metals. Considering the carbides presenting good strengthening effect, many researchers now add some carbon together with certain alloying elements into invar alloys. It seems that most of paper titles published on invar alloys don’t refer to carbon, even if the studied materials contain carbon. By convention, the paper title in our manuscript only refers to chromium, no carbon.

2, 10. Reply: We certainly expect to use an accurate equation to express variation in mechanical properties and coefficient of thermal expansion. However, the obtained data is not more, big errors will occur when doing curve fitting. Based on the trend lines for those parameters, we said the mechanical properties and CTE varied in a parabolic form.

3. Reply: We check the whole article, delete the semicolon and then start a new sentence. Please check details in the revised manuscript.

4. Reply: The manuscript novelty is that addition of chromium, together with certain amount of carbon, could strengthen the Fe36Ni invar alloy. Furthermore, a maximum strength of 644.4 MPa and a minimum coefficient of thermal expansion of 3.3×10-6 K-1 were obtained in the sample aged at 425oC for 7 h. The strength value is about 30% higher than that of the commercial invar alloy at a condition of remaining low thermal expansion. Those results were presented in the manuscript.

5. Reply: Perhaps the difference in locations results in different prices for the metal alloying elements, Cr (1.9$ per 100 g) is indeed cheaper than Mo (4.4$ per 100 g), V (31.3$ per l00 g), and Ti (5.4$ per 100g) in China. Considering the industrial application in the future for the studied invar alloy, we expect to adopt cheap alloying elements to decrease its cost. Therefore, the cheaper Cr element is given a priority.

6. Reply: We supplement the methodology of mechanical properties and CTE measurements in the Materials and Methods Section in the revised manuscript.

7. Reply: We check the whole article, and use “properties” substituting for “property”. Please check details in the revised manuscript.

8. Reply: The aging temperatures used in this study are 375~475oC. The hardness of the samples presents a sharp increase when aging in the temperature range from 375 to 425 oC. However, when the aging temperature is higher than 425 ℃, the hardness of the samples decreases with an increase in the aging temperature, as shown by solid line in Fig.1. Because the sample aged at 425 ℃ shows the maximum hardness, we studied effect of aging time on mechanical properties and thermal expansion at a fixed aging temperature of 425 oC in order to obtain a good strengthening effect.

9, 12. Reply: We changed the “EDX” into “EDS” in the article. Please check it in the revised manuscript.

11. Reply: The chemical compositions in Fig.4d, f and Fig.5d, e were expressed in at.%. Please check it in the revised manuscript.

Round  2

Reviewer 3 Report

The manuscript has been properly revised.

Reviewer 4 Report

The provided corrections and Authors explanations are satisfactory.